# ACER: Automatic Language Model Context Extension via Retrieval

## Abstract

Long-context modeling is one of the critical capabilities of language AI for digesting and reasoning over complex information pieces. In practice, long-context capabilities are typically built into a pre-trained language model (LM) through a carefully designed context extension stage, with the goal of producing generalist long-context capabilities. In our preliminary experiments, however, we discovered that the current open-weight generalist long-context models are still lacking in practical long-context processing tasks. While this means perfectly effective long-context modeling demands task-specific data, the cost can be prohibitive. In this paper, we draw inspiration from how humans process a large body of information: a lossy **retrieval** stage ranks a large set of documents while the reader ends up reading deeply only the top candidates. We build an **automatic** data synthesis pipeline that mimics this process using short-context LMs. The short-context LMs are further tuned using these self-generated data to obtain task-specific long-context capabilities. Similar to how pre-training learns from imperfect data, we hypothesize and further demonstrate that the short-context model can bootstrap over the synthetic data, outperforming not only long-context generalist models but also the retrieval and read pipeline used to synthesize the training data.

## 1 Introduction

The field of Artificial Intelligence (AI) and Natural Language Processing (NLP) have made substantial progress in building and teaching neural language models (LMs) to understand and generate language (Radford et al., 2019; Brown et al., 2020; OpenAI, 2023; Anthropic, 2023; 2024; Touvron et al., 2023a;b; MetaAI et al., 2024). Large-scale deep learning has enabled large LMs to learn from massive amounts of human-generated text (Radford et al., 2019; Brown et al., 2020). However, new challenges emerge as researchers consider building capabilities beyond those of humans. One popular example, also the focus of this paper, is understanding long-contexts of text (Dai et al., 2019; Su et al., 2024; Xiong et al., 2023). Despite the advancements in modeling (Dao et al., 2022; Su et al., 2024; Xiong et al., 2023), data keeps being lacking simply because humans no longer produce them naturally. Specifically, while longer contexts give more degrees of freedom in forming possible language sequences and long-interaction/complex-information tasks, the existing long texts are limited to organized and compact ones like novels and code (Fu et al., 2024).

A common methodology in building long-context LM is to incorporate a context extension phase into the model building cycle after the general pre-training phase and before the task-aware post-training phase (Xiong et al., 2023; Rozière et al., 2024; MetaAI et al., 2024). As the standard pre-training stage builds into the model the general capabilities over diverse text distributions, the long-context extension phase is designed with the hope to extend generalist capabilities further to long-context patterns. The subsequent post-training is supposed to activate and align these extra long-context capabilities to task instructions.

Despite the exciting promise of this context extension scheme, the limited portion of model training dedicated to long-context contradicts the aforementioned increasingly more complex space language and task patterns when the text gets longer (Xiong et al., 2023; MetaAI et al., 2024). In other words, using a context extension phase to cover all long-context understanding tasks can be mathematically intractable, and consequently, building a long-context generalist may not succeed. To investigate this, in this paper, we conduct experiments and demonstrate empirically that practical tasks like

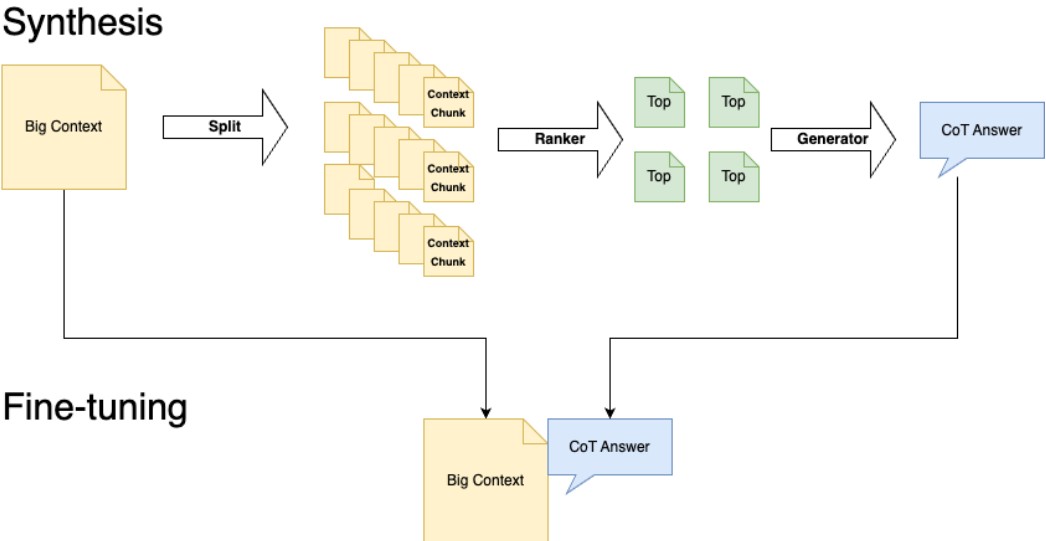

Figure 1: The full process of ACER involves a data synthesis stage and a fine-tuning stage. (top) The data synthesis stage splits and retrieves a set of relevant text chunks for a problem and use a short-context model to generate an answer with CoT (Wei et al., 2022). (bottom) The fine-tuning stage use the original long-context data and the synthetic CoT answer to fine-tune a long-context model.

long-context retrieval augmented generation can easily break existing long-context models. While creating domain-specific supervised training may help fix this. Unlike short-context training data that can be collected from everyday people (Kopf et al., 2023), long-context training can be much harder to curate. Reading long pieces of text is inherently hard for humans, which could make the annotation process not only costly but also very demanding and, therefore, less reliable.

In this paper, to alleviate this problem and provide an intermediate solution, we propose a new approach which **A**utomates **C**ontext **E**xtension via **R**etrieval (ACER; Figure 1). Overall, ACER is a two-stage method. We start with synthesizing *imperfect* data by combining retrieval with an LM *excels in short context*. In the subsequent stage, we will fine-tune a large LM to bootstrap over this data. ACER will start with some pairs of question and its long context, while no labeling is required. In the synthesis pipeline, the long context is broken into chunks and a retrieval model will score and rank the chunks. A small set of top-ranked chunks will be fed into the short-context LM to produce answer *with chain-of-thought* (CoT; Wei et al. (2022)) reasoning. Then, in the fine-tuning stage, we train the model using the context, question, and the CoT. On the other hand, the retrieval-based data synthesis process will be hidden from the model. We desire that the deep model will learn a generic long-context understanding function by fitting over the full context and the CoT reasoning. We hypothesize that the model may discover a better latent function that transcends the original ranking mechanism used in the first stage. This also shares some spirit with LM pre-training. Whereas typically pre-training bootstrap from existing human data, due to the apparent data scarcity, ACER will bootstrap from synthetic long-context data generated using retrieval.

Our experiments demonstrated the effectiveness of ACER. We found model trained with ACER without supervision can outperform contemporary generalist long-context models. It also outperforms its own retrieval-based answering pipeline if applied to the test sets.

## 2 ACER

In this section, we will give an overview of ACER. The ACER method consists of two major stages: 1) automatic data synthesis, and 2) self training, as illustrated in Figure 1. In this chapter, we will describe how each of these stages work.

## 2.1 Automatic Data Synthesis

Our data synthesis process combines a heuristic-based retrieval pipeline and a short-context generator. We use the following ingredients,

- **Prompts**: a set of prompts/problems, $\{p_1, p_2, .., p_N\}$, each of which consists of a pair of context and question $p_i = (c_i, q_i)$
- **Short-Context Ranker**: A ranker model $r(q, t) \to \mathbb{R}$ which takes a question $q$ and a piece of *short* text $t$. This ranker determines a relevance score corresponding to how helpful the text $t$ in answering the question $q$.
- **Short-Context Generator**: A generator model $g(x) \to a$ that takes some short prompt $x$ and returns an answer $a$. This will be an aligned instruction-tuned model like many existing contemporary LM.

We start the data synthesis process from the set of prompts. For one prompt $p_i = (c_i, q_i)$, we break the context into a set of text chunks $\{t_{i1}, t_{i2}, t_{i3}, ...\}$. The ranking model will assign chunk $t_{ij}$ a relevance score $s_{ij} = r(q_i, t_{ij})$. Based on these scores, we can produce a ranking of the text chunk indices using the estimated helpfulness, $[r_{i1}, r_{i2}, r_{i3}, ...]$. We collect the top $M$ ranked text chunks while making sure that their concatenation can still fit in the short generator. These "most helpful" chunks will together be fed into a generator to produce an answer,

$$\hat{a}_i = g(\texttt{INST} \circ q_i \circ t_{r_{i1}} \circ t_{r_{i2}} \circ .. \circ t_{r_{iM}}) \tag{1}$$

Here `INST` denotes an instruction to the model to elicit an explicit reasoning in a chain-of-thought like form. This makes sure that the model can describe how it compares the information in the provided text pieces to identify the most useful ones, as well as how it combines them to arrive at the final answer. This reasoning process will be used in the next stage to help train the model in the fine-tuning stage.

This data synthesis process leverages two critical capabilities in the original LM, relevance/usefulness analysis and understanding of short pieces of text. We combine them heuristically to build a surrogate long-context pipeline. Readers familiar with the concept of map-reduce may recognize our data synthesis as such a process.

## 2.2 Fine-tuning

During data synthesis, we produce for each prompt, a full document ranking, a small set of helpful documents and an answer. For fine-tuning, we discard the ranking and the document set and use only the generated answer, because we do not want our model to be exposed to and learn from the *lossy* retrieval pipeline. We fine-tune an LM $f_\theta$ to produce the CoT answer $\hat{a}_i$, i.e.,

$$\hat{\theta} = \text{argmin}_\theta \sum_i f(\hat{a}_i | (c_i, q_i)) \tag{2}$$

In this setup, we provide the model a lossless, unfiltered access to the full context. In addition, we pair it with an answer with extra training signals, a CoT describing an information extraction process of picking up useful pieces of information and sorting through. As with any other deep learning application, we desire that the large over-parametrized LM can learn during the optimization process to fit a long-context understanding function that will lead to similar reasoning and answer. To keep it simple, we fine-tune the model with teacher forcing using a log likelihood loss. Only the answer tokens participate in loss computation with the context token loss masked out during training.

## 3 Experimental Setup

### 3.1 Tasks

We consider two realistic tasks, long-context retrieval augmented generation (Jiang et al. (2024)) and long-context reading comprehension. We pick long-context RAG as our major evaluation since the task, while being very useful, has only very recently be carefully studied. This is more

aligned with our desired setup, distinct yet useful long context tasks with little supervision. We consider the following two datasets: **Natural Question (NQ)** (Kwiatkowski et al., 2019) and **Trivi-aQA (TQA)** (Joshi et al., 2017) We use Wikipedia as the knowledge source, and BGE (Xiao et al., 2024b) which is a dense retriever Karpukhin et al. (2020). We report the exact match (EM) metric.

For long-context reading comprehension, we used the **NarrativeQA** dataset Kočiský et al. (2018). The dataset has been out for several years at the time of writing and has since been a standard evaluation for long context. We expect many long-context models being trained on some form of its train set. Nevertheless, we still use it as reference to understand how our self-supervised approach compares to the supervised ones. We used the curated test set from LongBench Bai et al. (2023). We report the token-level F1 metric.

While these datasets have short gold answer, modern LMs tend to produce long answers. In order to evaluate EM or F1 scores, we employ an additional short answer extraction step by few-shot prompting a Llama-3-8B-Instruct model using the prompt introduced by Jiang et al. (2024).

## 3.2 DATA SYNTHESIS

For data synthesis, in order to keep it simple, we implement the ranker and the short context generator by prompting the same LM, Llama-3-8B-Instruct. This is the short-context version of the latest generation of Llama models (MetaAI et al., 2024). We use the 8B variant instead of the 70B or 405B to demonstrate the applicability of ACER in a cost efficient setup.

**Ranker Model**    We show the prompt of the ranker model in Figure 2. We instruct the model to read the question and context chunk (referred to as passage in the prompt) and *think step-by-step* to decide the helpfulness of the passage for answering the question. We formulate it as a multiple choice question where the model is instructed to choose from 5 options from the most a) "providing exact answer" to e) "not related".

---

**Ranker Prompt**

Let's think step by step: decide if the given question can be answered by the given passage.

Choose from the following options:
a) The passage provides the exact answer to the given question.
b) The passage provides a partial answer to the question.
c) The passage provides an answer to a similar but different question.
d) The passage is only tangentially related to the question.
e) The passage is not related to the question.

The last line of your answer should be: Answer: [a)/b)/c)/d)/e)].

Question: {question}

Passage: {passages}

---

Figure 2: Prompts given to the LM to produce relevance judgement.

**Generator Model**    We show the prompt for the generator model in Figure 3. The model is instructed to read the given passages and extract *several* pieces of potentially helpful information before it makes decision on what evidence to use and then use it to answer the question.

We use prompts in the corresponding dataset's training set. For RAG tasks, we obtain the context by using a dense retriever to retrieve 100 passages from Wikipedia. We use the DPR version of Wikipedia dump, where the documents are splited into chunks of 100 words. For NarrativeQA, the context is simply the original full book. We do not make any additional edit over it. We apply

---

**Generator Prompt**

Answer the following question by reading the given passages.

Question: {question}

Passages:
{passages}

Think carefully and start your answer by first extracting a few pieces of related information from the passages.
Then identify the most useful piece of information and use it to answer the question.
On the last line, end your response with a complete full sentence answer of the following format:
Answer: <complete answer>
Answer "No answer was found" if you cannot find the answer in the given passages.

---

Figure 3: Prompts given to the LM to produce the final CoT answer.

an addition data augmentation technique to the RAG data: we perform a sliding window shuffle of candidates passages. We use window size of 30 and a stride of 20. This is used to efficiently emulate a ranking noise where the absolute change in position decays with the change size. One copy of the noised data is used as well as one with original ranking.

For the creation of the final data, we reuse the generator prompt template (Figure 3) with the full context as question. We combine it with the generated answer using a chat template.

### 3.3 FINE-TUNING

We directly fine-tune the ACER models over long context without approximation. This section describes the training details as well as contemporary training technologies/techniques we adopted.

#### 3.3.1 IMPLEMENTATION

We train over the synthesized data for 1 epoch with a batch size of 128 examples. We use a max sequence length of 32k for RAG and 64k for reading comprehension, and therefore this amounts up to 4M tokens and 8M tokens batch, respectively. These lengths are picked to cover most of the test lengths while at the same time keeping training on our accelerators viable and efficient. At the test time, RoPE extrapolation (Su et al., 2024) is utilized for sequences of longer length. We use a $3e - 6$ learning rate with Adam optimizer (Kingma & Ba, 2015). For model initialization, we still require a model that has a sufficiently long context length. This can be achieved unsupervisedly by training on long-context corpora. To save cost, we borrow Llama-3-8B-ProLong-Base model (Gao et al., 2024)[1], an open model unsupervisedly context-extended from our data synthesizer model Llama-3-8B-Instruct (The authors made decision to build a base model from an instruction tuned version.)

#### 3.3.2 INFRASTRUCTURE

We train our models using JAX (Bradbury et al., 2018) on cloud TPUv4s (Jouppi et al., 2023) provided by TPU Research Cloud. These 4th generation TPUs are built with relatively small-size 32GB HBM per chip while interconnected with high bandwidth fibers. We therefore opt to partition computation and perform our training with 3D parallelism of data parallelism, sequence parallelism, and tensor parallelism. We train on 32 TPUv4 chips in a $(2, 4, 4)$ configuration mesh and map these axes to data, sequence, and tensor parallelism axes respectively. Optimizer states as well as the full-precision copy of model weights are kept fully sharded over the mesh, over the entire course of training. We manually shard model, optimizer, and critical activation in attention and feed-forward

---

[1] `princeton-nlp/Llama-3-8B-ProLong-512k-Base`

network. We use JAX's `shard_map` to shard flash attention (Dao et al., 2022; Dao, 2024) TPU kernels[2] across model axis by heads. The rest of the shardings are decided by the XLA SPMD compiler (Xu et al., 2021).

We compute loss only over the answer, with the long context receiving only implicit gradients. Taking advantage of this to save memory and flops, on top of using a loss mask, we (dynamically) slice the last layer of hidden states at each batch instance into a smaller answer tensor. Only this smaller tensor is passed to the LM head for out-projection and loss computation. In the backward pass, this will translate into a (dynamic) update of the gradient tensor.

## 3.4 COMPARED MODELS

For comparison, we consider models that are relatively open and closely related to the base model we use for ACER, Llama-3.1-Instruct. We acknowledge the effectiveness of proprietary models such as GPT4 OpenAI (2023). We do not include them here, because our focus is to evaluate a method of data synthesis and the models and data they use are not directly comparable.

Specifically, we compare ACER with long-context LMs fine-tuned on open data as well as those fine-tuned on closed data. Specifically, we consider the following,

- **Together-Llama-2-7B-32K-Instruct (Together, 2023)**: A model by TogetherAI by extending Llama2 to a 32k context size. It is fine-tuned on an open mixture of long-context question answering and summarization data.
- **Llama-3-8B-ProLong-512k-Instruct (Gao et al., 2024)**: This is a Llama3-8B-Instruct context extended on open long context dataset. It is then fine-tuned on UltraChat (Ding et al., 2023), a large fine-tuning dataset generated by GPT-4. The creator found it most helpful amongst open instruction-tuning datasets. This model closely relates to ours and share the same base model, offering a straightforward comparison.
- **Llama3-8B-Instruct (Truncation) (MetaAI et al., 2024)**: Llama3-8B-Instruct is a 8k context model pre-trained and instruction-tuned by MetaAI with closed data. In this setup, we truncate the input to fit it into the context.
- **Llama3-8B-Instruct (RAG) (MetaAI et al., 2024)**: This is Llama3-8B-Instruct reading a small set of input chunks generated with the same retrieval pipeline used in the ACER data synthesis.
- **Llama3.1-8B-Instruct (MetaAI et al., 2024)**: This is the second iteration of the Llama3 model by MetaAI. Its context is systematically extended by Meta GenAI team.
- **Mistral-Nemo-Instruct-2407 (MistralAI, 2024)**: A larger 12B model trained by MistralAI and Nvidia using closed data. It has a native 128k context length.

We perform inference with vLLM (Kwon et al., 2023) with greedy decoding. Models always read the full context and extrapolate when reading longer context than the original training-time length except in Llama3-8B-Instruct (Truncation).

## 4 EXPERIMENTAL RESULTS

In Table 1, we show the performance of the compared systems as well as ACER on the evaluation datasets. We see a general trend that ACER outperforms the compared systems with decent margins especially on the novel long-context RAG tasks. We observe a general trend that the closed data model performing better than models trained on open data. Specifically, the Together-Llama-2 model, which is based on the previous generation Llama model, significantly underperforms all other model. This is likely due to two facts that the model essentially stems from an earlier generation, and its shorter context requires it to do more extrapolation. Now, to remind our reader, the rest of the 8B models, all came from the same base model Llama-3-8B-Base. Despite this fact, we see vastly different performance. The ProLong model performs decently on NarrativeQA but falls behind on the RAG tasks. This is not surprising as the UltraChat supervision it uses relates more

---

[2] https://github.com/jax-ml/jax/blob/main/jax/experimental/pallas/ops/tpu/flash_attention.py

| Model | NQ (EM) | TQA (EM) | NarrativeQA (F1) |
|---|---|---|---|
| *Supervised Open Data* | | | |
| Together-Llama-2-7B-32K-Instruct | 0.172 | 0.299 | 0.013 |
| Llama-3-8B-ProLong-512k-Instruct | 0.260 | 0.457 | 0.150 |
| *Supervised Closed Data* | | | |
| Llama3-8B-Instruct (Truncation) | 0.388 | 0.567 | 0.076 |
| Llama3-8B-Instruct (RAG) | 0.408 | 0.611 | 0.161 |
| Llama3.1-8B-Instruct | 0.312 | 0.518 | **0.217** |
| Mistral-Nemo-Instruct-2407[12B] | 0.365 | 0.534 | 0.072 |
| *Self-Supervised* | | | |
| ACER (ours) | **0.446** | **0.648** | 0.189 |

Table 1: Performance comparison of ACER and baselines on Natural Question (NQ), TriviaQA (TQA), and NarrativeQA. The best performance on each dataset is boldfaced.

closely to standard-form question answering but can be very different from tasks in the wild like long-context RAG here. On the other hand, we see with the more involved long-context extension done by MetaAI, the Llama-3.1 model significantly outperforms Prolong. It actually also attains the best performance on NarrativeQA, likely because the data it uses may contain long-context QA data. Our ACER model is the second best on NarrativeQA, still out-performing all other systems, suggesting that our self-supervised method is still competitive when no specific supervised data is present. Nevertheless, ACER is the best-performing on the RAG datasets, again confirming the usefulness of our self-supervised method.

We do also note that the 12B Mistral-Nemo does not always show decisive advantage over the 8B models. This demonstrates further that the model size and capability advantage do not always translate into long-context performance. As we discussed before, long context means more task diversity and a model's task fitness becomes as critical; here we see the Mistral model performs decently on RAG but is lacking on Narrative QA. This again shows the usefulness of ACER which is able to self-supervisedly teach a model a new task.

When compared with Llama3-8B-Instruct (RAG), which uses Llama-3 with the ACER retrieval pipeline in a test-time ad-hoc manner, we still see a decent performance boost in ACER. This suggests that learning and bootstrapping from ACER generated data can transcend the original data synthesis pipeline, as we desired.

## 5 ANALYSIS

### 5.1 COMPARING LLAMA-3.1 AND ACER AT DIFFERENT CONTEXT SIZES

To get a more fine-grained understanding of how context extension affects each model, in this section, we compare models at a variety of context lengths. Specifically, we varies the number of retrieved passages fed into the model for generating the final answer. Recall that Llama-3.1 and ACER both derive from the same pre-trained Llama-3 base model using different context extension processes. In Figure 4, we plot the two models' performance against number of passages on both Natural Question and TriviaQA (1k subset to reduce inference cost.)

We observed a very interesting yet intuitive phenomenon. Here the two models perform very similarly when reading a small context. This means they possesses similar capability and alignment behavior at short context, as *sibling models*. However, as context increases, the performance keeps diverge. While ACER can keep digesting more useful information from the context, Llama-3.1 seemingly suffers from the longer contexts with performance decaying. This shows one other example that ACER achieved the goal we set for it and successfully **extended** the model's capability presented in a shorter context onto a much longer context.

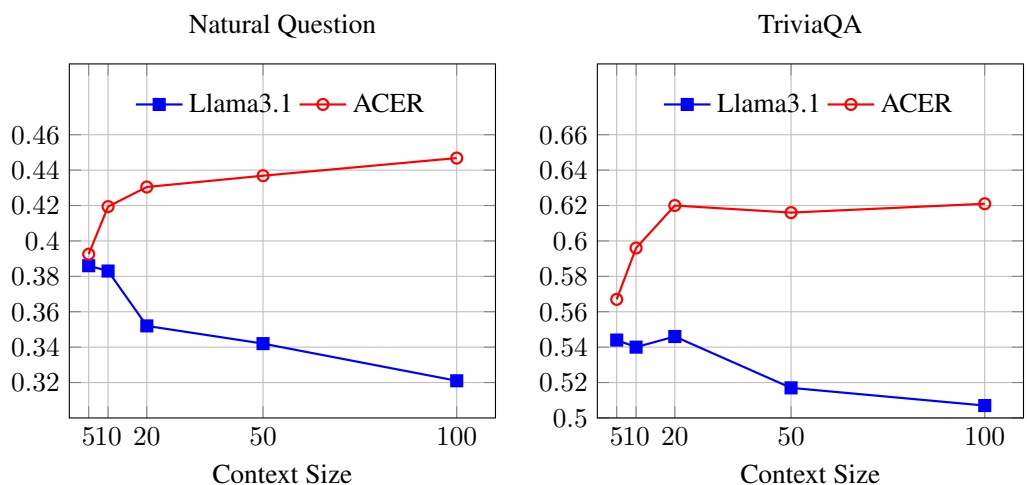

Figure 4: Performance comparison between Llama3.1 and ACER when reading different context sizes on the Natural Question and Trivia QA datasets.

## 5.2 USING AN UNSUPERVISED RETRIEVER

| Retriever | Model | NQ (EM) | TQA (EM) |
|---|---|---|---|
| BM25 | Llama3 (Truncation) | 0.260 | 0.542 |
| | Llama3 (RAG) | 0.354 | 0.576 |
| | Llama3.1 | 0.305 | 0.507 |
| | ACER | **0.383** | **0.632** |
| Dense | Llama3 (Truncation) | 0.388 | 0.567 |
| | Llama3 (RAG) | 0.408 | 0.611 |
| | Llama3.1 | 0.312 | 0.518 |
| | ACER | **0.446** | **0.648** |

Table 2: Performance of ACER and baselines with different base retrievers.

The previously discussed ACER pipeline for RAG tasks adopts a supervisedly trained dense retriever. In certain situations where even retrieval data is scarce, obtaining such a retriever may not even be possible. People instead need to fall back to the classical BM25 retrievers. In this section, we consider such a situation by running ACER and some of the baseline systems with BM25. This means the top-100 candidates will be different and of lower quality. In Table 2, we show the results of this BM25 setup compared with the original dense setup. While using BM25, all systems take a hit in performance because of lower-quality candidates, the general performance order of the systems remain the same. Our ACER method still outperforms the other systems by decent margins. In fact, when looking at the the absolute numbers, we can observe that ACER with BM25 attains close or even better performance than best-performing baseline systems. The results suggest that ACER is relatively agnostic to the retriever used and attains good performance to warm-start a system without requiring extra supervision.

## 6 RELATED WORKS

**Language Modeling** Recent advancements of large language models have shown strong language understanding ability and ace a wide range of natural language processing tasks. Based on the Transformer structure (Vaswani et al., 2017), LMs can be broadly categorized as decoder-only models (e.g., GPT (Radford et al., 2018; 2019)), encoder-only models (e.g., BERT (Devlin et al., 2019) and RoBERTa (Liu et al., 2019)), and encoder-decoder models (e.g., BART Lewis et al. (2020a)

and T5 Raffel et al. (2020)). Most recently, the large language models such as the GPT family (Brown et al., 2020; OpenAI, 2023), Gemini (GeminiTeam et al., 2024a;b), Llama Touvron et al. (2023a;b); MetaAI et al. (2024), and Claude Anthropic (2023; 2024) have attract great attention by achieving human-level performance on various tasks and showing structure-following ability. These models are typically trained in several stages, including pre-training on web-scale text data with auto-regressive language modeling, supervised fine-tuning on specific applications, and human preference alignment, where the last step plays the pivotal role to steer the LMs as a dialogue system to answer human's instruction and generate responses that are in desired quality and style, safe, and ethical. Some recent LM alignment methods include reinforcement learning from human feedback (RLHF) (Ouyang et al., 2022), proximal policy optimization (PPO) (Schulman et al., 2017), direct policy optimization (DPO) (Rafailov et al., 2023), and Kahneman-Tversky optimization (KTO) (Ethayarajh et al., 2024).

**Long Context Modeling** Transformer (Vaswani et al., 2017) is the foundation architecture for recent advancements in language models. However, its quadratic temporal and computation complexity to the sequence length poses great challenge to scale to long input sequence, and the lack of robust position embeddings also degrades the performance of long context understanding. Therefore, tremendous efforts are made to improve the long-context modeling of language models from different aspects, including more efficient attention mechanism for long context Beltagy et al. (2020); Kwon et al. (2023); Dao et al. (2022); Dao (2024); Xiao et al. (2024a), a (recurrent) internal or external memory bank (Dai et al., 2019; Wu et al., 2022a;b), and length extrapolation through positional encoding (Press et al., 2022; Su et al., 2024; Peng et al., 2024; Zhu et al., 2024). To evaluate the long-context modeling ability of LLMs, several synthetic benchmarks are proposed, such as Lost in the Middle (Liu et al., 2024), Needle in a Haystack (Kamradt, 2023), and Ruler (Hsieh et al., 2024). Because we are proposing new data synthesis method for long-context LM training, we do not include the synthetic benchmarks.

**Retrieval Augmented Generation** Retrieval-augmented generation, or RAG, enhances LLMs' ability in knowledge intensive tasks. Originated from open-domain question answering (Chen et al., 2017; Xiong et al., 2021b), RAG will enrich the prompt (i.e., a question) with additional relevant context retrieved from an external corpus to help a reader better answer the question (Lewis et al., 2020b; Guu et al., 2020; Izacard et al., 2024). The retrieval methods range from sparse features (Robertson & Zaragoza, 2009; Roberts et al., 2020) and deep-learning based dense representations (Karpukhin et al., 2020; Lewis et al., 2020b) to direct generation by LLMs (Sun et al., 2023; Yu et al., 2023). Recent studies have explored various ways to enhance RAG, such as better query understanding (Kim et al., 2023; Chan et al., 2024), a better retriever (Karpukhin et al., 2020; Xiong et al., 2021a; Yao et al., 2023), and a better reading model (Izacard & Grave, 2021; Cheng et al., 2021; Borgeaud et al., 2021). Specifically, Self-RAG (Asai et al., 2024) trains a critique model that reflects the retrieved passages and the generation to adaptively decide whether the retrieval is necessary. SuRe (Kim et al., 2024) proposes summarized retrieval which generate multiple summaries of retrieved context based on answer candidates. RAPTOR Sarthi et al. (2024) builds a hierarchical structure by iteratively cluster text chunks and generate summaries for the clusters, which can aggregate text chunks to better answer summarizing questions.

## 7 CONCLUSION

In this paper, we propose ACER, a method for automatically extending language model capabilities to longer contexts without using human generated supervision. ACER pivots through a retrieval pipeline to help a short-context model to heuristically process long pieces of text and produce a synthetic *imperfect* answer. We demonstrate that this model can then bootstrap over this synthetic answer to gain even stronger long context capabilities, often outperforms carefully built long-context generalist models. We believe ACER demonstrate an effective unsupervised approach to extend short context capabilities into longer contexts. It can be a useful tool to improve specific long-context task performance where little training data exists. Users only need to write a few prompts to run the pipeline. Broadly, we introduce automating capabilities extension onto longer context as a new research problem; future works may consider better data synthesis processes and paths to produce better extension results.

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
