# OpenReview forum: "ACER: Automatic Language Model Context Extension via Retrieval"
_ICLR.cc/2025/Conference — Submitted to ICLR 2025_

### Official Review · Reviewer_NhPs · 2024-10-27

**Soundness:** 3
**Presentation:** 3
**Contribution:** 2
**Rating:** 3
**Confidence:** 5

**Summary:**

The paper introduces a novel two-stage approach for enhancing the long-context capabilities of large language models without the need for human-generated supervision data. The first stage involves automatic data synthesis where a retrieval model and a short-context generator work in tandem to produce synthetic long-context data. The second stage is a fine-tuning process where a large LM is trained on the synthetic data to acquire long-context understanding abilities. The paper presents experiments on long-context retrieval augmented generation and long-context reading comprehension tasks, demonstrating the effectiveness of ACER.

**Strengths:**

(1) The paper presents an effective approach to extend the capabilities of LMs into longer contexts by leveraging retrieval mechanisms and synthetic data generation.
(2) The experimental results are positive, with ACER being tested on standard benchmarks like Natural Questions (NQ), TriviaQA (TQA), and NarrativeQA. The paper also provides a detailed comparison with other models, which strengthens the quality of the evaluation.
(3) The paper is well-structured and easy to follow.

**Weaknesses:**

(1) While the paper demonstrates the effectiveness of ACER on certain tasks, it does not explore the generalizability of the approach across different domains or longcontext evaluations. Additional experiments in this regard could strengthen the paper's claims.
(2) The idea is valid but lacks novelty. It is more like an engineering trick than a research idea.
(3) The article lacks some more in-depth analysis and ablation on experimental settings, such as the impact of chunk size and the potential relationship with context length.

**Questions:**

（1）Chunking strategies in long documents can disrupt the contextual flow and background information, resulting in some chunks containing incomplete data. This fragmentation may prompt LLMs to reference unrelated contexts or revert to their internal, parameterized knowledge, which can lead to inaccurate responses. How should ACER deal with this problem?
（2）Different question may need different (M) ranker text chunks, how can we set up a M for different problems?

---

### Official Review · Reviewer_WdtJ · 2024-10-29

**Soundness:** 3
**Presentation:** 2
**Contribution:** 3
**Rating:** 5
**Confidence:** 4

**Summary:**

This paper introduces a new method for extending the long-context capabilities of language models, called ACER (Automatic Context Extension via Retrieval). ACER enhances the long-context processing abilities of a short-context language model through a two-stage automated process.

Experimental results show that ACER outperforms other mainstream long-context models across multiple tasks and datasets, particularly excelling in long-context Retrieval Augmented Generation (RAG) tasks.

**Strengths:**

An effective pipeline for automatically generating RAG training data, achieving significant performance improvements even with just an 8B model as a data generator. The scores for downstream RAG tasks are also impressive.

**Weaknesses:**

1.	Most comparisons in the experiments are with Long-Context Models. I believe additional RAG strategies should be included for comparison.

2.	It would be helpful to see some comparative data statistics, such as a table showing Big Context length and CoT Answer length from Figure 1, as well as length comparisons in the experimental section.

3.	Figure 1 needs a higher-resolution version. Using a PDF image is recommended.

**Questions:**

Please check weaknesses section.

---

### Official Review · Reviewer_37rX · 2024-11-01

**Soundness:** 2
**Presentation:** 3
**Contribution:** 2
**Rating:** 5
**Confidence:** 3

**Summary:**

This paper addresses the limitations of current open-weight generalist long-context models in practical long-context processing tasks and proposes an innovative approach inspired by human information processing. By developing an automatic data synthesis pipeline that uses short-context language models to mimic human document retrieval and reading processes, the authors aim to enhance task-specific long-context capabilities. The study demonstrates that short-context LMs, when further tuned with self-generated data, can surpass both long-context generalist models and the foundational retrieval-read pipeline in effectiveness for tasks such as long-context retrieval augmented generation.

**Strengths:**

1. The paper is well-written.
2. The methodology is clear and effective.

**Weaknesses:**

1. The evaluation was conducted solely on long-context RAG tasks, where improvement is natural given the methodology. However, it was not assessed on more general long-context evaluation sets, such as LV-Eval and Needle in a Haystack.
2. The approach seems to be too simplistic and straightforward, lacking innovation and contribution.
3. Experiments are conducted on only one size and one type of language model.

**Questions:**

Please kindly address the weaknesses.

---

### Official Review · Reviewer_hFE4 · 2024-11-05

**Soundness:** 2
**Presentation:** 2
**Contribution:** 3
**Rating:** 5
**Confidence:** 3

**Summary:**

This paper explores the challenge of building language models with strong long-context processing capabilities, which are essential for tasks that require digesting and reasoning over complex information. The authors note that current generalist long-context models still fall short in practical applications, and that building task-specific long-context models can be prohibitively costly. To address this, the paper proposes an approach inspired by how humans process information - using a short-context language model to perform a lossy retrieval and ranking of a large document set, then fine-tuning that short-context model on the synthetically generated data to obtain task-specific long-context capabilities.

**Strengths:**

- The topic of long-context modeling is both compelling and critical, and this paper provides valuable new insights into addressing this task.

- The proposed method is well-conceived and alleviates the need for extensive resources for human-annotated data.

- The approach demonstrates the potential for practical application, making it a meaningful contribution to long-context modeling research.

**Weaknesses:**

- My primary concern with this paper is the limited evaluation. The experiments provide only a narrow comparison of long-context benchmarks, such as Infinibench [1] and LongBench [2]. Additionally, the paper misses several important approaches mentioned in paper [3] such as self-extend [4] and lm-infinite [5], but lacks a comparative analysis or discussion of these methods, which would strengthen the evaluation.

- There is also a lack of case studies and in-depth analysis of the model’s long-context modeling capabilities. Providing examples on synthetic data or offering a more detailed examination of attention patterns would help demonstrate that the model genuinely improves long-context handling, beyond the benchmark performance alone.

References:

[1] InfiniBench: Extending Long Context Evaluation Beyond 100K Tokens: Extending Long Context Evaluation Beyond 100K Tokens

[2] Longbench: A bilingual, multitask benchmark for long context understanding

[3] A controlled study on long context extension and generalization in llms

[4] Llm maybe longlm: Self-extend llm context window without tuning

[5] Lm-infinite: Simple on-the-fly length generalization for large language models

**Questions:**

- The capabilities of Llama-3-8b-it can sometimes be limited, and it may not consistently follow human instructions well in certain cases. How does this limitation impact the performance of your proposed method? Would using a larger backbone model (e.g., Llama-3.1-70b-it) provide greater benefits for the proposed approach?

---

### Meta-Review · Area_Chair_QC1C · 2024-12-21

**Metareview:**

Long-context language modeling remains challenging for open-weight models. This paper presents a new approach for long-context LM that leverages retrieval and data synthesis. This method first uses retrieval to identify a small number of relevant documents, generates CoT answers from these documents, and then fine-tunes a long-context LM on this new data with the CoT generation. Experiments conducted on long-context RAG and long-context QA benchmarks demonstrate notable performance improvements.

Strengths
- The paper tackles an important problem of long-context language modeling, and the new method that uses data synthesis is valuable (hFE4, WdtJ, NhPs).
- Empirical results are impressive (WdtJ, NhPs).
- The paper is well written (37rX, NhPs).

Weaknesses
- Limited evaluation, due to lack of evaluation benchmarks and baselines (hFE4, 37rX, NhPs, WdtJ)
    - The paper mainly shows improvements on long-context RAG tasks, where improvements are expected as it exactly matches how synthetic data was generated (37rX). Whether it improves general long-context LM benchmarks that are beyond RAG remains underexplored (37rX, NhPs).
    - Lack of comparison to an approach that uses RAG at test time (WdtJ).
- Insufficient technical contribution (37rX, NhPs)
- Lack of details/analysis, such as examples of the generated data, data statistics, and analysis on attention scores (hFE4, WdtJ).

**Additional Comments On Reviewer Discussion:**

No author responses provided

---

### Decision · Program_Chairs · 2025-01-22

Reject